# Parent Education for Responding to and Supporting Youth with Suicidal Thoughts (PERSYST): An Evaluation of an Online Gatekeeper Training Program with Australian Parents

**DOI:** 10.3390/ijerph19095025

**Published:** 2022-04-20

**Authors:** Samuel McKay, Sadhbh J. Byrne, Alison Clarke, Michelle Lamblin, Maria Veresova, Jo Robinson

**Affiliations:** 1Orygen, Parkville, VIC 3052, Australia; byrnes83@tcd.ie (S.J.B.); alison.clarke@ucdconnect.ie (A.C.); michelle.lamblin@orygen.org.au (M.L.); maria.veresova@orygen.org.au (M.V.); jo.robinson@orygen.org.au (J.R.); 2Centre for Youth Mental Health, The University of Melbourne, Parkville, VIC 3010, Australia; 3Centre for Global Health, Trinity College Dublin, D02 K104 Dublin, Ireland

**Keywords:** suicide prevention, parents, carers, gatekeeper training, online training

## Abstract

The gatekeeper training of parents is a promising approach for suicide prevention in young people, but little research has addressed the effectiveness of such training, especially using online delivery. This study aimed to evaluate the efficacy and acceptability of the delivery of an online suicide prevention training program, LivingWorks Start, to improve the capacity of parents to support young people at risk of suicide. The participants were 127 parents of young people aged 12–25 who completed the LivingWorks Start training and consented to participate in the evaluation. The participants completed online surveys before, after, and 3 months after training. The participants showed increases in perceived self-efficacy and formal help-seeking intentions but no change in suicide stigma. Suicide literacy also increased, but only at the three-month follow-up. Most parents found the training acceptable, and did not find it upsetting. Prior mental health, suicide-related experiences, and pre-participation vulnerability were not predictive of finding the training distressing. Overall, the findings show that online gatekeeper training for parents can be beneficial, and is rarely associated with distress.

## 1. Introduction

Suicide is the second leading cause of death for young people aged 15–29 [1]. Suicide attempts, self-harm, and suicidal feelings are also common among young people, and are associated with a wide range of heightened risks [2]. Parents have been identified by both young people and those who work with young people as a key target group for youth mental health and suicide prevention research [3,4]. Parents are well-placed to observe suicide warning signs, monitor risk, encourage alternative coping strategies, provide emotional support, and facilitate engagement with mental health services [5], and as such may be an important point of early intervention [6]. However, research indicates that parents’ knowledge about suicide and confidence in their ability to intervene is lacking, e.g., [7]. A small number of suicide prevention gatekeeper programs have been tested internationally; they show that the empowerment of parents through suicide education may be potentially efficacious for the improvement of knowledge, attitudes and help-seeking intentions in parents [8]. In turn, increased knowledge or ‘literacy’ of suicide has been found to predict the provision of more appropriate support to people experiencing distress [9]. Nevertheless, engaging parents in training programs can be challenging.

Poor rates of recruitment and retention are a universal challenge in parent-focused research [10]. As such, the exploration of facilitators of and barriers to participation is crucial in order to optimise parent engagement and maximise the subsequent benefits for young people. Online delivery may help to reduce barriers associated with face-to-face delivery methods [11], and has been found to be an effective delivery method for suicide prevention training in studies with clinicians [12] and school staff [13]. However, to date, evaluations of online programs with parents are lacking.

In this paper, we evaluate an online suicide alertness training program called LivingWorks Start. The pedagogical principles underpinning LivingWorks Start are based on the safeTALK program developed by the same organisation, and have been found to be effective among Australian secondary school students [14] and a variety of adult populations, e.g., [15,16]. However, until now, no systematic evaluation of the LivingWorks Start program has been undertaken.

The present evaluation aimed to examine the efficacy and acceptability of delivering LivingWorks Start training to parents of young people aged 12–25 years. The key outcome measures for the training program included the perceived self-efficacy to support a child experiencing suicidal thoughts and feelings, help-seeking intentions for a child experiencing suicidal thoughts, knowledge of suicide in the form of suicide literacy, and stigmatising attitudes toward people who carry out suicide. Each of these variables has been commonly utilised in gatekeeper training evaluations and broader suicide prevention literature, e.g., [8,9,13,14,16]. Additionally, a measure of program acceptability was included in order to capture participants’ experiences and perspectives on the program after completion. Past experiences of suicidal thoughts and behavior, as well as mental health problems, were also captured in order to control for their potential impact on finding the training upsetting. Finally, post-participation distress was measured in order to assess whether parents found participating in the research distressing. The following hypotheses were tested:Completion of the LivingWorks Start training program by parents will be associated with improved self-efficacy to support a young person experiencing suicidal thoughts and feelings, increased intentions to seek help for their child, improved suicide literacy, and reduced suicide stigma.Parents’ scores on each variable will remain stable from post-intervention to follow-up.LivingWorks Start training is not associated with increased distress for parents.LivingWorks Start training is acceptable to parents (i.e., enjoyable and worthwhile).

Additionally, the research included an exploratory component addressing the perceived barriers to and facilitators of parents taking part in the current study.

## 2. Materials and Methods

### 2.1. Study Design

A pre-test/post-test design with a three-month follow-up period was employed. The participants were assessed two weeks before (Time One) and immediately after the training (Time Two), and then three months following (Time Three), using a specifically designed online survey. The participants were sent a weekly reminder to begin the program for three weeks before being withdrawn from the study. If the participants did not complete any of the three questionnaires within three days of it being sent, the research assistant would send one reminder to complete it. The participants who completed all three questionnaires received a gift voucher worth $25.

### 2.2. Participants

The participants were parents or primary caregivers of young people aged 12 to 25 years, living in Victoria, Australia, and they were recruited via schools, social media, and other community groups. All of the parents who completed the Time One questionnaire and provided informed consent were eligible to participate in the training and the remaining two time points. The exclusion criterion was an inability to converse in or read English, as this may preclude full engagement in the LivingWorks Start intervention or appropriate completion of the measures. Four-hundred and forty parents consented to participate and completed the baseline questionnaire. However, 301 participants did not complete the LivingWorks Start program. Furthermore, of the 139 participants who completed LivingWorks Start, 12 participants did not complete either the Time Two or Time Three surveys, and were thus removed. The final sample comprised 127 participants at Time One, 121 at Time Two, and 104 at Time Three. The attrition across time occurred due to not all of the participants completing all of the follow up surveys when they were invited to do so via email.

### 2.3. Intervention

LivingWorks Start is an online, 90-minute, self-paced, interactive community training programme designed for those aged 13 years and older. Through the use of video and text-based training materials, the program aims to increase awareness of suicide, to develop basic skills at intervening with someone who is considering suicide, and to connect the suicidal person with suicide first aid help and community resources which are relevant to mental health.

### 2.4. Measures

#### 2.4.1. Demographic Information

The participants were asked to indicate their gender, their number of children aged 12–25, the age and gender of one of their children within this age range, their relationship to their child, their family structure, whether or not they identified as Aboriginal or as a Torres Strait Islander, their country of birth, the main language spoken at home, their education level, their current employment status, their relationship status, whether they work in healthcare, their sexual orientation, their current living situation, and who they live with. The participants were also asked to indicate how they found out about the present study.

#### 2.4.2. Post-Participation Distress

Two scales developed by Yeater et al. [17] were used to assess participation-induced distress in absolute terms (i.e., whether participation was distressing) and in relative terms (i.e., whether participation was more distressing than everyday life events) at Time Three. The absolute measure included six items. The participants reported the option that best represented their opinion on a 7-point Likert scale ranging from one (Strongly Disagree) to seven (Strongly Agree). The scores were summed, with a potential range of 6–42, with higher scores representing greater absolute distress occurring as a result of the study. Relative distress was assessed with five items, in which the participants compared the experience of the study with everyday life events on a scale from one (“The event described would be much worse than this study”) to seven (“This study was much worse than the event described”). The scores were summed with a potential range of 5–35, with higher scores indicating greater relative distress experienced through participation in the study. Cronbach’s α for the absolute measure of distress was 0.76, while it was 0.92 for the relative measure.

#### 2.4.3. Perceived Self-Efficacy

The parents’ perceived self-efficacy to engage in activities to prevent, or assist their child in managing, a suicidal crisis was assessed using nine questions developed by Czyz et al. [7] on a 10-point Likert scale ranging from zero (Not at all confident) to 10 (Completely confident). The scores were summed to produce a total score, with higher scores reflecting greater perceived self-efficacy, with a possible range of 0–90. Cronbach’s α in the present sample was 0.92.

#### 2.4.4. Help-Seeking

Help-seeking intentions were assessed using an adapted version of the General Help-Seeking Questionnaire [18], which includes 10 items on a 7-point Likert scale ranging from one (Extremely Unlikely) to seven (Extremely Likely). Following the methodology of Wilson [19], the scale was split into informal sources (e.g., a partner, friend, parent, other family member/relative) and formal sources (e.g., a mental health professional, phone helpline, Doctor/GP). The scores for formal and informal help seeking were summed to produce a total score, with higher scores reflecting increased help-seeking intentions for the relevant source, with a possible range of 4–28 for informal and 3–21 for formal sources. Cronbach’s α in the present sample was 0.64 for informal sources and 0.62 for formal sources.

#### 2.4.5. Suicide Stigma

Suicide stigma was assessed using the short form of the Stigma of Suicide Scale (SOSS) [20], which includes 16 items rated on a 5-point Likert scale ranging from one (Strongly disagree) to five (Strongly agree). In the current study, only the 8 items of stigma subscale were used, and the item scores were summed to create a total suicide stigma score. The possible range of the measure was 8–40, with higher scores indicating greater suicide stigma. Cronbach’s α in the present sample was 0.93.

#### 2.4.6. Suicide Literacy

The impact of the program on suicide literacy was assessed using the short form of the Literacy of Suicide Scale (LOSS) [21]. This includes 12 items answered in a true or false format. The number of correct answers provided by the participants was converted to an overall percentage, with higher scores indicating greater literacy of suicide.

#### 2.4.7. History of Suicidal Ideation and Behaviour

Family history of psychological problems and suicide was assessed by the questionnaire in Appendix A. The questionnaire covered the following: the parent’s personal history of mental health problems, suicidal ideation and self-harm; knowing somebody who performed suicide; supporting somebody who is engaged in suicidal ideation or behaviours (including their child); and previous suicide prevention training.

#### 2.4.8. Program Acceptability

Acceptability was assessed at Time Two only. Six items assessed (a) the participants’ enjoyment of the training, (b) whether or not they would recommend it to other parents, (c) whether or not they found the training upsetting, (d) whether the training met their expectations, and (e) whether there was anything they would change about the delivery of the program. The participants could choose to respond either ‘yes’ or ‘no’ to each question except the question regarding their expectations, for which the options were ‘worse than expected’, ‘about the same as expected’ or ‘better than expected’. The questions were scored by calculating the proportion of participants endorsing each possible response.

#### 2.4.9. Perceived Barriers and Facilitators

The participants were asked three open-ended questions: “What do you think puts parents off taking part in research, such as the present study?”, “What do you think encourages parents to take part in research, such as the present study?”, and “What made you decide to take part in the present study?” The main idea(s) present in every response were extracted, then grouped into broader concepts that represented similar ideas, for example, “wanting to learn more about suicide prevention”. The frequency and percentage of each concept occurring was then recorded. This process was performed for each of the three questions separately.

### 2.5. Data Analysis

The data were analysed using SPSS version 27 (IBM, Armonk, NY, USA). A linear mixed-effects model was used to examine the changes in each continuous outcome measure (suicide literacy, self-efficacy, suicide stigma, and help-seeking) over time. This approach was chosen because linear mixed-effects models are robust regarding missing data (e.g., a participant can be missing data at one time point but still be included in the model) and do not assume the independence of observations, reducing the risk of biased standard errors [22]. The model was parameterised such that coefficients refer to mean scores at Time One, Time Two, and Time Three (by including terms for all three time points in the model but excluding the intercept term). The results contain an unadjusted analysis where the time period variable is the only predictor, and an adjusted analysis that contains gender, number of children they had aged 12–25, age and gender of one of the children, relationship to their child, family structure, whether or not they identified as Aboriginal or Torres Strait Islander, whether or not they were born in Australia, whether or not English was the main language spoken at home, level of education, current employment status, relationship status, whether they work in healthcare, sexual orientation, current living situation, and who they live with as covariates. Post hoc testing was used to assess whether there was a change over time, whether Time Two scores differed from Time One scores, and whether Time Three scored differed from Time Two scores. Logistic regression analysis was conducted to assess whether prior experiences related to mental health problems, self-harm, or suicide contributed to participants finding the program upsetting.

## 3. Results

### 3.1. Participant and Demographic Characteristics of the Sample

The mean age of the sample was 48.33 (SD = 6.15) and the reported gender was 89.9% female, 8.9% male, and 1.6% other genders. The remaining demographic characteristics are presented in Appendix B. There was no difference in age (*p* = 0.455), gender (*p* = 0.600), number of children (*p* = 0.672), relationship status (*p* = 0.264), or employment status (*p* = 0.169) for those who completed only Time One and those who completed the subsequent time points (e.g., Time Two and/or Time Three).

### 3.2. Self-Efficacy, Help-Seeking, Suicide Stigma, and Suicide Literacy

In both the adjusted and unadjusted analyses (Table 1), self-efficacy, formal help-seeking all improved significantly from Time One to Time Two. However, only formal help-seeking did not decrease significantly from Time Two to Time Three, with self-efficacy showing a small but significant decreases. Suicide literacy did not significantly increase from Time One to Time Two, but improved from Time Two to Time Three. Informal help-seeking and suicide stigma showed no change across time.

### 3.3. Program Acceptability

The acceptability questions were completed by 121 participants. Most reported finding LivingWorks Start enjoyable (N = 112; 92.6%), that they would recommend it to others (N = 119; 98.3%), and that the program was not upsetting (N = 98; 81%). Most of the participants said that the program exceeded (N = 75; 62.0%) or met (N = 44, 36.4%) their expectations, while a very small portion found the training worse than expected (N = 2; 1.7%). Logistic regression showed that previous suicidal thoughts (OR = 0.45, 95% CI 0.14–1.48, *p* = 0.189), a prior mental health diagnosis (OR = 1.40, 95% CI 0.42–3.49, *p* = 0.732), past self-harm behaviour (OR = 0.89, 95% CI 0.22–3.57, *p* = 0.870), having a child ever engage in suicide-related behaviours (OR = 1.06, 95% CI 0.38–2.99, *p* = 0.907), having known one or more people who died by suicide (OR = 1.22, 95% CI 0.40–3.66, *p* = 0.729), and previously completing suicide training (OR = 0.80, 95% CI 0.27–2.42, *p* = 0.698) were not associated with finding the training upsetting.

### 3.4. Post-Participation Distress

The post-program distress measures were completed by 102 participants. The participants’ mean score on the absolute measure of post-participation distress was 12.13 (SD = 5.20), while the mean score for the relative measure was 13.76 (SD = 7.61), both of which show that, on average, the participants did not find the training distressing.

### 3.5. Perceived Barriers and Facilitators

The most common perceived barriers to participating in the study were a lack of time (N = 91; 71.7%), finding the topic too difficult or uncomfortable (N = 16; 12.6%), and privacy/confidentiality concerns (N = 14; 11.0%). The most frequently named facilitators for parent participation in suicide research, generally, were wanting to learn information or skills about the topic of suicide (N = 29; 22.8%), relevance to the self or family (e.g., knowing someone who died by suicide, child experiencing suicidality or mental health concerns, etc.; N = 22; 17.3%), and personal interest in the topic (N = 20; 15.7%). The most common perceived facilitators for participation in this specific study were: direct relevance or experience (N = 40; 31.5%), wanting to learn information and skills (N = 32; 25.2%), and personal interest in the topic of suicide (N = 19; 15.0%).

## 4. Discussion

### 4.1. Key Findings

This is the first study to assess an online suicide-specific education program, LivingWorks Start, with Australian parents. Overall, the participants showed increases in perceived self-efficacy to prevent, or assist their child in managing, a suicidal crisis, formal help-seeking intentions for their child experiencing suicidal thoughts, and reduced suicide stigma. These changes followed the intervention (Time Two) and were maintained at the three-month follow up (Time Three), except for suicide stigma, which returned to the baseline at Time Three. Suicide literacy also increased during the study, but this change occurred between Time Two and Time Three, and thus may not be related to the program. The participants reported no change in suicide stigma or informal help-seeking intentions for their child experiencing suicidal thoughts.

Most of the participants reported that they enjoyed the training, would recommend it to others, and that it met or exceeded their expectations. Most did not find the training distressing. The parents reported choosing to participate because they wanted to know more about the topic of suicide, because it was relevant to their child, or because of personal interest. The reported barriers to participation were time constraints, finding the topic difficult or uncomfortable, and confidentiality concerns. In combination, the findings suggest that online suicide prevention training for parents is potentially effective, acceptable, and meets parents’ needs.

### 4.2. Implications

This study shows that gatekeeper training delivered online can enhance parents’ self-efficacy. This is a notable finding, as self-efficacy has been found to be the strongest predictor of adults’ intentions to help individuals with mental health difficulties [23], and of university students’ intentions to intervene with a suicidal individual [24]. Higher parental self-efficacy is also predictive of increased adherence to support recommendations for their children after discharge from emergency departments for suicide [25]. However, it is important to remain cognizant that participants’ responses in this study are reflective of their stated confidence to act, rather than their actual behaviour.

Furthermore, although parents may feel more confident to intervene, it is important that their responses are perceived as appropriate by their children. Other Australian research [26] has found that parents’ responses to suicidal ideation disclosure were perceived as the least helpful of all informal/non-professional sources of support. This is noteworthy given that the current study found no change in parents’ suicide stigma. These findings suggest that training may need to further target stigma in order to ensure that parents do not intervene in a stigmatising or inappropriate manner, as this may increase the risk of their children perceiving their support as unhelpful. Taken together, the results suggest that it is crucial to examine both the impact of training on the actual behaviour of parents and how this behaviour is perceived by their children. While the present study did not have the opportunity to investigate actual behaviour, we recognise that this as an important direction for future research.

The current study also shows that gatekeeper training for parents enhances formal but not informal help-seeking intentions, which aligns with previous findings on interventions targeting help-seeking for mental health problems more generally [27]. Such a finding is also not unsurprising, given that the evaluated training emphasises connecting individuals with formal support services after the identification of suicide risk. Nevertheless, the documented increase is still notable, given that the scores would already be considered high (17.99 out of a possible 21 points) at the beginning of the study.

Unexpectedly, the training did not immediately enhance suicide literacy, but literacy increased 3-months post training. One possible explanation for this finding is that the training did not provide the already quite suicide-literate participants (86.8 out of a possible 100 points) with novel information but instead encouraged them to further engage with suicide prevention information, enhancing literacy in the longer term. However, further research is required in order to better understand this process.

More broadly, this study highlights the potential benefits of involving parents in youth suicide prevention efforts, and recognising their role as gatekeepers or ‘gateway providers’ [28]. This suggests that such programs could be delivered more widely as part of state or national suicide prevention activities. Further research should explore the best practice in involving parents in care for young people experiencing suicidal ideation and/or engaging in self-harm, as highlighted by recent policy recommendations [29,30].

The perceived barriers and facilitators to participation reported in the present study are in line with those presented in previous research [10]. This implies that some of these barriers and facilitators are not specific to suicide research, and thus necessitate considerations of how methodological designs can be improved to increase parent participation. While barriers such as time constraints can be minimised through the use of online interventions, the Internet may introduce additional barriers such as potential privacy concerns. The confrontational or uncomfortable nature of the topic is a barrier that may be specific to suicide prevention research, and should be considered when developing recruitment strategies for such studies. It is also noteworthy that 306 of the participants who initially signed up for the study and completed the baseline survey did not go on to complete the training or subsequent surveys. While we cannot determine the reasons for this attrition, it is possible that an aspect of the current study design may have presented a barrier to completion or contributed to participant drop-out.

### 4.3. Limitations

The findings of this study should be considered in light of several limitations. First, the short-term follow-up period and absence of a control group means we cannot be sure that changes in self-efficacy, help-seeking intentions or suicide literacy were retained long term or changed as a result of the training, as the effects could be a result of repeated testing. Second, the vast majority of the sample (89.9%) were female, which, while typical of this type of research [31], warrants caution when generalising these results to parents of all genders. Third, our sample may have been affected by the social media recruiting bias. For instance, samples recruited from social media have been noted to have an overrepresentation of Caucasian women [32], as well as higher education levels [33]. These characteristics were also highly prevalent in our current sample. It is also possible that self-selection bias played a role, with individuals who have lower levels of suicide literacy and higher levels of negative attitudes towards suicide being less likely to choose to participate. Fourth, there was only one Aboriginal and no Torres Strait islander participants, and only a small proportion of the sample was born outside of Australia, thus limiting the generalisability of the findings. Further research is needed to confirm the current findings and overcome the above-mentioned limitations by evaluating the LivingWorks Start program in a large-scale randomised control trial with more diverse samples. It is also important to note that the COVID-19 pandemic may have influenced the progression of this study in terms of making online support more accessible or desired at this time. COVID-19 likely created a greater perceived need for mental health support, and caused parents to seek more resources, or to experience their own struggles. The pandemic also limited our ability to recruit through schools, as we originally intended. It remains to be seen what the recruitment, participation, and nature of responses would be like for an analogous study that is not as affected by the pandemic. Finally, it would be beneficial to undertake comparison studies between in-person and digital gatekeeper training in order to assess whether the different formats are more or less efficacious or acceptable.

## 5. Conclusions

This study provides preliminary evidence for the benefits of online gatekeeper training aimed at parents of young people. Such training may increase parents’ self-efficacy and help-seeking intentions for their children if they experience a suicidal crisis. Parents reported that the training was acceptable and most did not find it distressing. The online format appears to address barriers to research participation related to accessibility, but presents other challenges, such as privacy concerns. Future research using a randomised control design would facilitate stronger conclusions on the benefits of the program. Nevertheless, the current findings show that online suicide prevention training for parents is a promising avenue for future suicide prevention work.

## Figures and Tables

**Table 1 ijerph-19-05025-t001:** Mean scores for outcome variables at each time point based on unadjusted and adjusted multilevel linear regression models.

	Unadjusted			Adjusted		
	Mean	95% CI ^a^	*p* Value ^b^	Mean	95% CI	*p* Value ^b^
**Self-Efficacy**						
Time One	60.12	56.85–66.39	-	60.67	57.52–63.83	-
Time Two	79.59	77.86–81.31	**<0.001**	79.89	78.19–81.59	**<0.001**
Time Three	77.58	75.78–79.59	**0.033**	77.86	75.79–79.94	**0.035**
**Formal Help-Seeking**						
Time One	17.99	17.43–18.55	-	18.03	17.46–18.60	-
Time Two	19.29	18.85–19.72	**<0.001**	19.32	18.89–19.76	**<0.001**
Time Three	19.03	18.53–19.54	0.325	19.03	18.53–19.53	0.254
**Informal Help-Seeking**						
Time One	16.75	15.71–17.78	-	16.87	15.87–17.87	-
Time Two	17.34	16.29–18.39	0.153	17.38	16.35–18.40	0.219
Time Three	17.79	16.74–18.85	0.415	17.79	16.76–18.81	0.467
**Suicide Stigma**						
Time One	9.92	9.15–10.68	-	9.84	9.07–10.61	-
Time Two	9.61	8.80–10.42	**0.568**	9.54	8.71–10.36	**0.582**
Time Three	9.63	8.87–10.39	**0.958**	9.54	8.78–10.31	**0.998**
**Suicide Literacy**						
Time One	0.86	0.84–0.88	-	0.86	0.84–0.88	-
Time Two	0.87	0.85–0.89	0.306	0.87	0.86–0.89	0.309
Time Three	0.90	0.88–0.91	**0.014**	0.90	0.88–0.91	**0.018**

^a^ Confidence interval; ^b^ tests the hypotheses that Time One scores differ from Time Two scores, and that Time Two scores differ from Time Three scores. *p* Values in bold represent significant effects at *p* < 0.05.

## Data Availability

Due to the nature of this research, the participants of this study did not agree for their data to be shared publicly; as such, supporting data are not available.

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
