# Peer review of "Parent Education for Responding to and Supporting Youth with Suicidal Thoughts (PERSYST): An Evaluation of an Online Gatekeeper Training Program with Australian Parents"

_ijerph, 2022, doi:10.3390/ijerph19095025_

Round 1
Reviewer 1 Report
The study deals with an important area of ​​prevention of loss of long life years among young people and hence its importance. This is done through the LivingWorks program, which aims to increase suicide awareness, develop basic intervention skills for those identified as having suicidal intentions or thoughts, and connect them to first-aid suicide prevention and community mental health resources. The study analyzes the sample with its limitations that are quite extensively detailed in the chapter on limitations in the study and the main one in which there is no control group to compare the responsiveness and effectiveness of an online parent training program compared to a frontal parent training program. Numerous studies exist on face-to-face parental intervention programs. I would expect to compare them versus it is necessary to compare the results of the online model in similar populations and go a little beyond the analysis of this specific sample for comparison versus results in parent-to-face guidance in other studies. But this study was limited to the sample only. But since this is a preliminary study by definition it is less possible to be careful about it.
The researchers are aware that the vast majority of the sample are women and only 8.9% are men. Undoubtedly this is a built-in filter with higher sensitivity of women in the field and responsiveness accordingly. In my opinion, the study also filtered out a lower socioeconomic population, otherwise I do not find a reason why $ 25 is an incentive to participate and not the desire to know more about an area that may cost children's lives. Lack of time can also be a filter for low socioeconomic sampling. The responsiveness to participating in the program after the initial intentional reduction from sample 440 to 127 in the first time, in one time, in the second time 121 and in the third time 104 indicates something about the rate of decline in the interest of the study participants. However, no significant demographic difference was found among the dropouts.
Undoubtedly, the online technology that became accessible and useful during the COVID-19 epidemic that greatly helped the program's success in research and will probably remain a convenient and economical option for training and content transfer in the future. Therefore, the technology should not be seen as a temporary condition but it will stay with us for the future and even improve.
It was also worth comparing the demographic distributions to the population registry and using this to find out in which audiences the response to the study was high and in which low. For example in men was expected to have a sampling potential of 50%. Why only 8.9%? What would have encouraged more men to participate? Even a comparison to the 440 that expressed correctness can be interesting.
There is a point that the researchers had to examine more and that is to examine the intervention plan at the stage of suicidal thoughts and when there is no distress in a concrete threat. It is therefore also possible unexpectedly, the training did not immediately improve suicide literacy but only over a longer period of time.
Cronbach’s α in the present sample was .64 for informal sources and .62 for formal sources. Where is Cronbach's α score for each dimension? Why were items that conflicted with answers not deducted and removed to improve the index?
Reviewer 2 Report
This is a manuscript that addresses a relevant topic such as the prevention of suicide in youth with the collaboration of parents.
The introduction is adequate for the purpose of the manuscript.
Materials and method
It is not clear why a study design with a control group was not used.
It is necessary to report Cronbach's α of the post-participation distress scales.
The paragraph on the suicide stigma scale appears to be flawed in that it mentions suicide literacy rather than attitudes toward suicide (lines 133-134).
It is necessary to describe how the instrument was scored on the acceptability of the program.
Likewise, a further description of the procedure to analyze the responses on Perceived Barriers and Facilitators is necessary.
Before data analysis, it is necessary to describe the instrument application procedure in three moments of the Pre-Post-Follow up evaluations. It is advisable to homogenize the way of naming these three evaluation moments pre-post-follow up or Time one, time two, time three.
It is necessary to explain why they decided to use the linear mixed-effects model instead of another statistical test for repeated measures.
Why was logistic regression analysis not used to assess the effect of previous experiences of suicide and mental health problems on the 3 main instruments in the study (suicide literacy, self-efficacy, suicide stigma, and help-seeking)? Personal experiences could influence these measurements.
It is necessary to include the ethical considerations of the study.
Results
It is not clear why the n of participants is different at the times of evaluation. This discrepancy needs to be clarified.
The phrase “However, only those who met the eligibility criteria, completed the Living-Works Start program, and completed at least one of the follow-up evaluations were included in the current results” is confusing. Specifically, it is not clear “completed at least one of the follow-up evaluations”, when it was supposedly a single follow-up evaluation three months after receiving the training.
The wording of the sentence “for those who completed the program evaluation and those who only completed the first survey” (lines 187-188) is confusing. Was the First survey the same as the Time one or pre-test assessment? Were those who completed the program evaluation the same who participated in post-test assessment or time two?
Was the acceptability of the program applied in the post-test evaluation or time two?
Post-program distress was applied in the follow-up evaluation – time three? Why is the n of participants different from those reported in the pre-post follow-up evaluations?
Regarding Perceived barriers and facilitators, it is necessary to report the number of participants and the time of their application. Report the n in addition to the percentage.
The discussion and conclusion are pertinent to the findings.
Reviewer 3 Report
First question, in order of importance, arises from this consideration: the control group is absent.How can the observed change be attributed to the effect of training without a control group?
Further requests (with minor relevance):
- Introduction is very short, add more elements to explain the variables included in the study
- Move line 18-182 from Results to Participants paragraph and add more details about participant selection, how many excluded for eligibility criteria? How many excluded for incomplete data? How many dropouts have there been?
- There is an incongruence between the number of items, their Likert scale from 1 to 7 and the potential range of absolute and relative distress scores. Check them.
- Add Cronbach’s alpha for two measures of distress.
- Line 135 I think you spelled “suicide literacy” instead of “suicide stigma”. Check it.
- Add Cronbach’s alpha for suicide literacy.
- Add Informal Help-Seeking results in Table 1.
- I see in table A1 that 37% of the participants had only one child aged 12-25.What did those with multiple children indicate in the questions "Gender of one child", "Age of one child" and "Relationship to one child"?
- I suggest more caution with terms/expressions as “directly attributed to the program” or “demonstrated” (lines 238-239) in discussion.
- I fully endorse the Limitations and Conclusions paragraphs, which also provide an answer to my first question
Round 2
Reviewer 2 Report
I appreciate the modifications made by the authors to the manuscript.
An error, the paragraphs in the limitations section appear in bold.
Best regards